# Bis-Cyclic Guanidine Heterocyclic Peptidomimetics as Opioid Ligands with Mixed μ-, κ- and δ-Opioid Receptor Interactions: A Potential Approach to Novel Analgesics

**DOI:** 10.3390/ijms23179623

**Published:** 2022-08-25

**Authors:** Jay P. McLaughlin, Ramanjaneyulu Rayala, Ashley J. Bunnell, Mukund P. Tantak, Shainnel O. Eans, Khadija Nefzi, Michelle L. Ganno, Colette T. Dooley, Adel Nefzi

**Affiliations:** 1Department of Pharmacodynamics, University of Florida, Gainesville, FL 32610, USA; 2Center for Translational Science, Florida International University, Port St. Lucie, FL 34987, USA; 3Torrey Pines Institute for Molecular Studies, 11350 SW Village Parkway, Port St. Lucie, FL 34987, USA

**Keywords:** heterocyclic peptidomimetics, combinatorial chemistry, solid-phase synthesis, opioid receptors, multifunctional opioid agonist, antinociception

## Abstract

The design and development of analgesics with mixed-opioid receptor interactions has been reported to decrease side effects, minimizing respiratory depression and reinforcing properties to generate safer analgesic therapeutics. We synthesized bis-cyclic guanidine heterocyclic peptidomimetics from reduced tripeptides. In vitro screening with radioligand competition binding assays demonstrated variable affinity for the mu-opioid receptor (MOR), delta-opioid receptor (DOR), and kappa-opioid receptor (KOR) across the series, with compound **1968-22** displaying good affinity for all three receptors. Central intracerebroventricular (i.c.v.) administration of **1968-22** produced dose-dependent, opioid receptor-mediated antinociception in the mouse 55 °C warm-water tail-withdrawal assay, and **1968-22** also produced significant antinociception up to 80 min after oral administration (10 mg/kg, p.o.). Compound **1968-22** was detected in the brain 5 min after intravenous administration and was shown to be stable in the blood for at least 30 min. Central administration of **1968-22** did not produce significant respiratory depression, locomotor effects or conditioned place preference or aversion. The data suggest these bis-cyclic guanidine heterocyclic peptidomimetics with multifunctional opioid receptor activity may hold potential as new analgesics with fewer liabilities of use.

## 1. Introduction

Pain is a major health problem that substantially reduces quality of life and imparts high health costs and economic loss to society. The urgent medical need for novel and safe analgesics with high efficacy has fueled intense research. The pharmacological differences produced by activation of the three opioid receptors encourage the search for compounds that produce analgesia without the deleterious side effects of morphine or other clinically used opioid analgesics. Despite more than 60 years of effort to understand the pharmacological intricacies of the opioid receptor family [1,2,3,4,5,6], most current clinically-used opioid analgesics are mu-opioid receptor (MOR) agonists and retain serious side effects [4,7,8,9]. Analgesic effects can also be mediated through other members of the opioid receptor family, such as the kappa-opioid receptor (KOR) and delta-opioid receptor (DOR), and the related nociceptin/orphanin FQ peptide receptor (NOP receptor) [9,10,11]. There is a growing interest for the generation of multifunctional ligands: novel opioid analgesics that can bind with high affinity and activate multiple receptors, opioid or otherwise [10,12,13,14,15,16,17,18,19]. Opioids possessing activity at more than one receptor have been suggested to produce potent analgesia with reduced undesirable effects [20]. Examples of multifunctional opioid receptors ligands targeting multiple opioid receptors with improved analgesic effects and reduced side effects include the bifunctional MOR/NOP receptor agonists SR16835/AT-202 and SR14150/AT-200, which have been shown to simultaneously activate both NOP and mu-opioid receptors to produce a wider therapeutic window and alleviate severe pain conditions [16,21]. Another example is the multifunctional opioid agonist dihydroetorphine, which binds the three opioid receptors with nanomolar affinity or better [22,23], and produces analgesia up to 12,000-fold more potent than morphine [16,22,23]. Similarly, the efficacy of LP1 in the treatment of persistent pain conditions has been ascribed to its dual MOR/DOR agonist activity [14,24,25]. Seeking promising analgesics with mixed-opioid agonism, we recently reported the synthesis of a series of diazaheterocyclics from reduced tripeptides and their in vitro screening with radioligand competition binding assays [13]. We identified compounds with varying affinity for the three opioid receptors in vitro, with subsequent antinociception in vivo of the diimidazodiazepine **14** (Figure 1) being attributed to agonist activity at all three opioid receptors (although with a preference for DOR [13]). Compound **14** showed limited tolerance when administered i.c.v. and tested in an acute tolerance paradigm [13], consistent with evidence suggesting multifunctional opioids may produce less tolerance. Moreover, upon central administration of doses producing maximal antinociception, compound **14** did not produce effects in place-conditioning or locomotor assays. Together, these results suggest possible advantages for mixed-activity opioid agonists in the face of clinical liabilities produced by morphine and other receptor-selective opioid agonists [13]. Structurally diverse ligands are also useful probes for studying the mechanisms of analgesic efficacy, addiction, and tolerance, let alone to develop effective new analgesics or other therapeutics [2,4,10,26,27].

Extending our earlier studies synthesizing novel agents from resin-bound peptides, we report presently the synthesis and in vitro and in vivo characterization of opioid activity of a series of bis-cyclic guanidine compounds derived from resin-bound reduced tripeptides: the **TPI-1968**-series. We have previously used our approach for the exhaustive reduction of resin-bound amides to generate resin-bound polyamines, used in turn as templates for the solid-phase synthesis of highly diverse heterocyclic libraries [28,29,30,31,32,33,34,35,36,37,38]. The amino acid side chains in polypeptide hormones, neurotransmitters, growth factors, substrates, antigens, and other bioactive peptides have been established as important pharmacophores for receptor/acceptor binding and signal transduction, and are determinant for the identification of new analgesics [39,40,41,42,43,44,45]. Previous reports from our lab show the utility of the transformation of linear peptides to peptidomimetics and heterocyclic peptidomimetics for the identification of several low-molecular-weight compounds in opioid receptor binding assays as potential pain modulators [13,31,45,46,47,48,49]. Likewise, we and others have validated cyclic guanidine analogs as potential pain modulators. Guanidinium and amino imidazolinium derivatives of *N*-4-piperydyl-propanamides were shown to have high affinity for mu-opioid receptors [50]. Interestingly, guanidines were used to improve the physicochemical characteristics of the putative endogenous MOR agonist EM-1, subsequently achieving greater blood–brain barrier (BBB) permeation [50]. Our group reported the screening and deconvolution of a bicyclic guanidine library derived from resin-bound reduced tripeptides led to the identification of an individual bicyclic guanidine with good affinity (IC_50_ value = 37 nM) for the KOR [51]. We also identified novel potent, low-liability antinociceptive pyrrolidine-tethered bis-cyclic guanidines from the direct in vivo screening of a large mixture-based combinatorial library [49]. Two compounds so identified demonstrated a dose-dependent antinociception three to five-times greater than morphine that was antagonized by mu- and/or kappa-opioid receptor-selective antagonists, and yet did not produce significant respiratory depression, hyperlocomotion, or conditioned place preference [49].

Presently, the **TPI-1968** series compounds were subjected to initial pharmacological screening in vitro for opioid receptor affinity with radioligand competition binding assays. The antinociceptive activity of six opioid ligands so identified were then characterized in vivo with the mouse 55 °C warm-water tail-withdrawal test after (i.c.v.) administration, and for lead bis-cyclic guanidine **1968-22**, oral (p.o.) administration, and opioid receptor agonist and antagonist profiles were assessed. Bioavailability of **1968-22** in both the blood and brain was evaluated up to 30 min after intravenous (i.v.) administration. Finally, the effect on acute antinociceptive tolerance, coordinated locomotor activity, respiration, and place-conditioning preference was investigated to evaluate potential liabilities of use.

## 2. Results

We successfully used resin-bound polyamines for the generation of a large number of heterocyclic compounds [30,35,52]. We performed the parallel synthesis of 64 bis-cyclic guanidine’s analogs of the highly active permethylated YYF that was previously identified (at a screening concentration of 0.5 nM) as a MOR antagonist [31]. As outlined in Figure 1, chiral polyamines were generated in our laboratory following exhaustive reduction of resin-bound peptides [30,35]. The reaction conditions were successfully optimized and all chemical modifications of the functionalized amino acid side chains were studied. As was reported in numerous publications, typical reaction conditions consist of the treatment of resin-bound polyamides with the complex BH3-THF [28,30]. The generated resin-bound tetraamines containing three secondary amines and one terminal primary amine (Figure 1) were treated with cyanogen bromide to promote cleavage from the resin of the corresponding bis-cyclic guanidines in the **TPI-1968** series.

Based on previous studies from our laboratory [31,45,48,53,54,55,56,57] and other groups [39,43,44,58], tyrosine and aromatic residues were universally found in the expansive family of opioid peptide and peptidomimetic ligands. As analogs of phenyl and hydroxyphenyl groups, we also used the non-natural amino acid 2,6-dimethyl-*L*-tyrosine (Dmt), which has become one of the most widely used amino acids in the synthesis of opioid peptides and pseudopeptides [43,59,60,61]. The residue Dmt plays a crucial role in the proliferation of opioid peptides with high receptor affinity (with a K_i_ value equal to or less than 1 nM) and potent bioactivity [59]. Thus, selecting Phe, Tyr, Ala and Dmt for the first position of diversity (R^1^), and Phe, Tyr, Leu and Dmt for the positions of diversities (R^2^) and (R^3^), we performed the parallel synthesis of 64 (4 × 4 × 4) bis-cyclic guanidines. All the compounds were purified and tested in competitive radioreceptor assay for opioid activity in mu, delta, and kappa-opioid receptors (see Appendix A). Twenty-six bicyclic guanidines with activities (K_i_ values) ranging between 7 and 93 nM for the MOR were identified (Appendix A), with eight compounds demonstrating MOR affinity of 30 nM or less (Figure 2). Some of the compounds also demonstrated modest affinity for kappa and/or delta-receptors, notably 1968-19 (with a K_i_ value of 66 nM for the KOR), and to a lesser degree 1968-22, with moderate K_i_ values of 218 nM and 265 nM for the KOR and DOR, respectively.

### 2.1. In Vivo Pharmacological Evaluation

Selected **1968**-series compounds were initially evaluated for their antinociceptive activity in the 55 °C warm-water tail-withdrawal assay in C57BL/6J mice following i.c.v. administration (Figure 2). All of the compounds produced time- and dose-dependent antinociception (each *p* < 0.05; Two-way RM ANOVA; see Table 1). On balance, peak antinociception was produced 20 min after i.c.v administration for all compounds except **1968-4** and the reference compound morphine, which both displayed peak antinociception 30 min after administration. At maximum doses tested, the duration of significant antinociception (*p* < 0.05; Dunnett’s post hoc test) varied from 60 min (**1968-4**, **-45** and **-47**) to 70 min (**1968-16**, **-22** and **-24**). All of the compounds exhibited full and potent antinociception except for compounds **1968-4** and **1968-47** (Figure 2A), which produced approximately 60% and 72% antinociception, respectively, at the highest dose tested (30 nmol). Morphine and the **1968**-series compounds all displayed varying potencies, with **1968-22** proving significantly different (F_(5,214)_ = 62.8; *p* < 0.0001, nonlinear regression modeling; Figure 2A). While the potencies of most **1968** compounds were comparable to that of morphine (Table 1), the ED_50_ (and 95% confidence interval) value of **1968-22** was 0.011 (0.005–0.026) nmol, i.c.v., over 264-fold more potent than morphine. Moreover, both **1968-4** and **1968-22** demonstrated significant antinociception (F_(11,77)_ = 4.72, *p* < 0.0001; Two-way RM ANOVA) in the tail-withdrawal assay after oral (p.o.) administration of a 10 mg/kg dose (Figure 2B). While antinociception induced by **1968-4** was brief, **1968-22** produced significant antinociception up to 80 min after oral administration (*p* < 0.05, Sidak’s multiple comparison test).

### 2.2. Opioid Receptor Selectivity of Antinociception Induced by 1968-Series Compounds

Pretreatment of mice with the opioid antagonist naloxone (10 mg/kg, s.c., −15 min) was used to assess receptor contribution to the observed antinociception induced by maximal tested doses of each **1968**-series compound tested. Naloxone pretreatment (15 min) demonstrated opioid receptor mediation of all antinociception, as this opioid antagonist significantly reduced the antinociceptive effects of all five analogs (*F*_(5,84)_ = 7.12, *p* < 0.0001, two-way ANOVA with Sidak’s post hoc test; Figure 3A). As **1968-16** and **1968-22** proved most susceptible to antagonism by naloxone, the individual opioid receptor contributions to their observed antinociception were then determined by pretreating the mice with the receptor-selective antagonists β-FNA (10 mg/kg, i.p., −24 h; for the MOR), nor-BNI (10 mg/kg, i.p., −24 h; for the KOR), or naltrindole (20 mg/kg., i.p., −20 min; for the DOR), before administering **1968-16** (at 30 nmol, i.c.v.) or **1968-22** (at 0.3 nmol, i.c.v.; Figure 3B). Treatment with these antagonists significantly reduced antinociception produced by either compounds (*F*_(3,56)_ = 84.56, *p* < 0.0001, two-way ANOVA). β-FNA, nor-BNI, and naltrindole all significantly antagonized the antinociception of each compound (*p* < 0.0001, Tukey’s post hoc test). Collectively, these results suggest that all three opioid receptors contributed to the antinociception produced by these two bicyclic guanidine compounds.

### 2.3. Determination of Opioid Receptor-Mediated Selective Antagonist Activity of 1968-22

Potential opioid antagonist activity of **1968-22** was next assessed, with mice pretreated with **1968-22** administered either the KOR-selective agonist U50,488 (10 mg/kg, i.p.), the MOR-preferring agonist morphine (10 mg/kg, i.p.), or the DOR-selective agonist SNC-80 (100 nmol, i.c.v.). A 2.5 h pretreatment with **1968-22** (0.3 nmol, i.c.v.) did not have an effect on morphine- or U50,488-induced antinociception, but significantly antagonized SNC-80 (*F*_(2,55)_ = 6.27, *p* = 0.004, two-way ANOVA with Sidak’s post hoc test; Figure 4A). **1968-22** antagonism of SNC-80 proved to be dose-dependent (*F*_(4,39)_ = 9.43, *p* < 0.0001, one-way ANOVA with Dunnett’s post hoc test), with a maximal antagonism observed with a 3 nmol, i.c.v. pretreatment (Figure 4B). Collectively, these results suggest that **1968-22** possesses multifunctional pan-opioid agonist/DOR antagonist activity.

### 2.4. Assessment of Pharmacokinetic Stability and CNS Penetration of Bis-Cyclic Guanidine 1968-22

The most potent of the tested compounds, **1968-22**, was selected for further study. To assess the stability of **1968-22** in plasma and evaluate CNS penetration, blood and perfused brains were collected from mice 5, 15, and 30 min after (i.v.) administration of the bis-cyclic guanidine (10 mg/kg, i.v.). **1968-22** was readily detected by LC−MS/MS analysis in serum samples within 5 min of administration and remained detectable 30 min later (Figure 5), indicating stability in mouse plasma. In contrast, negligible levels of **1968-22** were detected in perfused brain samples at any time point, further suggesting poor penetration across the blood–brain barrier.

### 2.5. In Vivo Assessment of Opioid-Related Liabilities of 1968-22

Given its potency and multifunctional pharmacological activity, **1968-22** was subsequently assessed for several potential liabilities produced by opioid agonists, specifically analgesic tolerance, hyperlocomotion, respiratory depression, and conditioned place preference.

### 2.6. Assessment of Acute Antinociceptive Tolerance 

**1968-22** was tested in a model of acute antinociceptive tolerance with repeated dosing (at 0 and 8 h, 0.001–100 nmol, i.c.v.) of morphine or **1968-22** [62,63]. The development of acute antinociceptive tolerance was assessed by pretreating with the ED_50_ i.c.v. dose of the test compound, followed 8 h later by treatment with one of a range of graded doses; antinociceptive tolerance was indicated by a significant increase in the ED_50_ value compared to the value observed in naïve animals. As expected, morphine demonstrated acute antinociceptive tolerance, with a 7.71-fold rightward shift in the dose-response curve of the second dose administered (to 23.8 (17.0–31.6) nmol, i.c.v.; Figure 6). **1968-22** demonstrated greater acute antinociceptive tolerance than morphine, with a 14.6-fold rightward shift in its second dose-response curve (to 0.18 (0.09–0.32) nmol, i.c.v).

### 2.7. Evaluation of Respiratory and Spontaneous Locomotor Effects of 1968-22

Mice were tested in the Comprehensive Laboratory Animal Monitoring System (CLAMS) to assess the effect of **1968-22** on spontaneous respiration rates and locomotor activity (Figure 7). As expected, over 2 h the positive control morphine (100 nmol, i.c.v.) produced significant, time-dependent respiratory depression compared to vehicle (0–40 min; *F*_(5,85)_ = 2.74, *p* = 0.02, two-way RM ANOVA with Dunnett’s multiple comparison post hoc test; Figure 7A), whereas treatment with **1968-22** (100 nmol, i.c.v.) resulted in significant (*F*_(20,255)_ = 2.75, *p* = 0.0001, two-way RM ANOVA), dose- and time-dependent *increases* in respiration rates (20–40 and 60–80 min; *p* < 0.05, Dunnett’s post hoc test; Figure 7A). Lower doses of **1968-22** (0.3, 1 or 10 nmol, i.c.v.) did not have any significant effect on respiration compared to vehicle (Figure 7A). Of interest, a 100 nmol i.c.v. dose of both morphine (*F*_(5,85)_ = 6.60, *p* < 0.0001, two-way RM ANOVA) and **1968-22** (*F*_(20,255)_ = 3.52, *p* < 0.0001, two-way RM ANOVA) had significant effects on ambulation compared to vehicle, demonstrating elevated ambulations at several time points (Figure 7B). In contrast, lower doses of **1968-22** were without significant effect (1 or 10 nmol), or produced brief reductions in ambulations (0.3 nmol, i.c.v., 0–20 min; *p* = 0.03, Dunnett’s post hoc test).

### 2.8. Evaluation of Potential Reinforcing or Aversive Properties of 1968-22

**1968-22** was assessed with a conditioned place preference assay (Figure 8). After two days of place conditioning, morphine (10 mg/kg, i.p.) induced significant place preference for the morphine-paired chamber, whereas U50,488 (10 mg/kg, i.p.) induced a significant conditioned place avoidance (*F*_(4,80)_ = 7.37, *p* < 0.0001; two-way ANOVA with Sidak’s multiple comparison post hoc test; Figure 8). However, mice place conditioned with **1968-22** at doses of 0.1, 1 or 10 nmol, i.c.v. demonstrated no significant preference or aversion for their respective drug-paired chamber.

## 3. Discussion

Extending our previous studies, the current **TPI-****1968** series of bis-cyclic guanidine compounds derived from resin-bound reduced tripeptides demonstrates the potential of multifunctional opioid agonists in the ongoing search for efficacious analgesics with reduced side effects. Central (i.c.v.) administration of the lead compound **1968-22** produced opioid receptor-mediated antinociception without antinociceptive tolerance, locomotor hyperactivity or impairment, or conditioned place preference. Moreover, **1968-22** demonstrated antinociception lasting at least 120 min after oral (p.o.) administration in the tail-withdrawal assay. LC−MS/MS analysis detected significant levels of **1968-22** in blood, but not brain, samples analyzed up to 30 min after i.v. administration, strongly suggesting compound stability in plasma with negligible CNS penetration.

Compound **1968-22** demonstrates a profile of multifunctional opioid agonism in vivo, displaying potent dose- and time-dependent antinociception blocked fully by pretreatment with the opioid antagonist, naloxone, or significantly by individual opioid receptor-selective antagonists. These results are consistent with previous demonstrations where multifunctional opioids produce a potent antinociception attributed to synergistic agonist activity at multiple opioid receptors, such as the synergistic MOR/KOR agonist activity observed with the cyclic tetrapeptide *cyclo*[Pro-Sar-Phe-D-Phe] [64,65]. Notably, activation of multiple opioid receptors to produce synergistic analgesia has been demonstrated repeatedly through studies co-administrating MOR, KOR and DOR agonists [66,67], including a study co-administering the MOR agonist DAMGO and either the KOR agonist U50,488 or DOR agonist DPDPE that generated dose-dependent, synergistic increases in antinociception [68,69]. The three opioid receptors are expressed in the peripheral sensory neurons, spinal cord and supraspinal regions that regulate nociception [69], supporting the interpretation that activation of all three types of opioid receptors could produce additive or synergistic antinociception of greater potency, as observed presently.

A major liability of clinically used MOR-selective agonists is that they cause significant tolerance upon repeated administration. Consistent with this and earlier demonstrations [63], a second administration of morphine presently produced acute antinociceptive tolerance. Compounds possessing multifunctional opioid agonism have been reported to produce less tolerance [13]. Our lead compound **1968-22** demonstrated mixed opioid receptor agonist activity, producing antinociception through the MOR, KOR, and DOR (though with perhaps a higher preference for DOR). However, in contrast with the absence of acute antinociception tolerance attributed to the mixed-opioid agonist activity for diimidazodiazepine 14 [13] or other mixed-opioid agonists [70,71,72,73,74,75] in other models, **1968-22** showed significant tolerance when administered i.c.v. and tested in an acute tolerance paradigm [63].

However, **1968-22** presently lacked many of the characteristic liabilities of established MOR agonists, including respiratory depression or conditioned place preference indicative of abuse potential. Multifunction opioids have been previously reported to display fewer side effects and clinical liabilities than observed by receptor-selective agonists [20]. For instance, the multifunctional KOR partial agonist/MOR antagonist nalbuphine produces analgesia with negligible respiratory effects and minimal abuse potential in both human and animal studies [76,77], and a number of multifunctional MOR agonists/DOR antagonists have demonstrated potent antinociception in animal studies with significantly reduced tolerance and physical dependence [17]. Admittedly, these examples draw on a combination of opioid agonist and antagonist activity. The demonstrated DOR antagonism of **1968-22** might have provided this benefit, although this pharmacological activity was mostly observed at higher doses than required for antinociceptive activity. Alternatively, the contrasting side effects of MOR and KOR agonism identified in **1968-22** may further counteract one another’s undesired effects, improving the therapeutic index. For instance, while MOR-selective agonists induce hyperlocomotion and are reinforcing, and KOR-selective agonists such as U50,488 both suppress coordinated locomotor activity [78] and cause conditioned place aversion [79,80], multifunctional opioids such as *cyclo*[Pro-Sar-Phe-D-Phe] demonstrate no such effects; behaviors attributed to the offsetting contributions of MOR and KOR agonism [62,65]. Notably, a handful of reports have shown KOR agonists to increase respiration, reversing MOR agonist-mediated respiratory depression [81,82]. Similar to these observations, the absence of respiratory depression induced by *cyclo*[Pro-Sar-Phe-D-Phe] was attributed to contributions of KOR agonist activity to the multifunctional MOR/KOR agonism [65]. These effects were confirmed by robust respiratory depression when the multifunctional cyclic tetrapeptide was administered to KOR KO mice [65]. Together with the present findings, the combination of multifunctional agonist activity at the three opioid receptors seems to minimize typical side effects relative to the gains in antinociceptive efficacy, as demonstrated by **1968-22**. It should be acknowledged that **1968-22** may display liabilities in different assays such as rodent self-administration, or different testing conditions such as peripheral administration of higher doses. Still, the current results suggest future work exploring the ratio of opioid receptor agonism required to modulate and/or mitigate liabilities of receptor-selective agonists, such as with opioid receptor knockout mice, would be of value to better understand this relationship.

Finally, while our data suggests **1968-22** is orally active, it must be noted that limited supply of **1968-22** precluded expanded studies utilizing peripheral administration. Such testing is of interest, as **1968-22** was detectable in plasma but not the brain after i.v. administration. These results raise the interpretation that **1968-22** is poorly or impermeable to the blood–brain barrier. Notably, our diimidazodiazepine **14** [13] also showed minimal brain penetration after oral administration [13], and was subsequently suggested to be peripherally restricted. While the current multifunctional data were deliberately based on central (i.c.v.) administration to avoid potential confounds of biological barrier permeability, additional studies of this property with both pharmacokinetic and pharmacological methods such as systematic application of naloxone methiodide [13] are warranted to address this question.

In conclusion, our synthesis of bis-cyclic guanidine heterocyclic peptidomimetics from reduced tripeptides yielded compounds with a variable range of affinities for the three opioid receptors, with compound **1968-22** displaying good affinity for MOR, DOR and KOR. Central intracerebroventricular (i.c.v.) administration of **1968-22** produced dose-dependent, opioid receptor-mediated antinociception in the mouse 55 °C warm-water tail-withdrawal assay, but did not produce locomotor effects, or conditioned place preference or aversion. The data suggest these bis-cyclic guanidine heterocyclic peptidomimetics with mixed-opioid receptor activity may hold potential as new analgesics with fewer liabilities of use.

## 4. Materials and Methods

### 4.1. Chemistry Synthesis

All the reagents, amino acids, and solvents were commercial grade. LC-MS (ESI) traces were recorded on samples with concentrations of 1 mg/mL in 50:50 MeCN/water at both 214 nm and 254 nm using a reverse phase Vydac column with a gradient of 5 to 95% formic acid in MeCN. The purity of the crude samples was estimated based on the UV traces recorded (214 nm and 254 nm). Hydrofluoric acid cleaves were performed in specially equipped and ventilated hoods with full personal protective equipment. All synthesized compounds were purified by RP-HPLC (Shimadzu Prominence HPLC system, Kyoto, Japan).

The identity and purity of the compounds were verified by Shimatzu HPLC and mass spectra under the following conditions: column, Phenomenex Luna 150 × 21.20 mm, 5 micron, C18; mobile phase, (A) H_2_O (+0.1% Formic acid)/(B) MeCN (+0.1% Formic acid), and 3 gradient methods used based on compound hydrophobicity (2% B to 20% B, 11 min; 25% B to 45% B, 31 min; 45% B to 65% B, 21 min); flow rate, 12 mL/min; detection, UV 214 nm. All chirality data were generated from the corresponding amino acids. Under our reaction conditions, the epimerization is minimized to less than 5% [31,35,83,84].

### 4.2. Typical Procedure for the Individual Synthesis of Bis-Cyclic Guanidine

All individual **1968** series compounds were synthesized in parallel following the strategy outlined in Figure 1 [85]. 100 mg of p-methylbenzdrylamine (pMBHA) resin per compound (1.2 mmol/g, 100–200 mesh) was sealed in a mesh “tea-bag”, [86] neutralized with 5% diisopropylethylamine (DIEA) in dichloromethane (DCM), and subsequently swelled with additional DCM washes. Boc-amino acid (6 equiv., 0.1 M in DMF) was coupled to MBHA resin using the classical coupling reagents, DIC and HOBt (6 equiv. each), for 2 h at room temperature, followed by washes with DMF and DCM (3 times). The Boc group was deprotected using 55% TFA in DCM for 30 min, followed by neutralization with 5% DIEA in DCM. Subsequent couplings of Boc-amino acid and deprotections of the Boc group were performed to yield a tripeptide. The completeness of the couplings was monitored by the ninhydrin test.

Exhaustive reduction of the amide bonds: the resin-bound tripeptides were treated in 50 mL glass conical tubes under nitrogen with borane–THF complex (1 M, 40 equiv.) at 65 °C for 72 h, followed by decantation of the reaction solution and quenching with MeOH [30]. Following washes with THF (4 times), the resin was treated with piperidine at 65 °C for 20 h to disproportionate the borane complexes. Following decantation of the piperidine–borane solution, the resin packet was washed with DMF (4 times), DCM (4 times) and MeOH (2 times), and dried.

The generated resin-bound tetra-amine was treated with CNBr (3 equiv., 0.02 M in DCM) under anhydrous conditions for 3 h, followed by washes with DCM (3 times). The desired compound was cleaved from the resin in the presence of anhydrous HF/anisole (95:5) at 0 °C for 1.5 h, and the cleaved product was extracted with 95% acetic acid in H_2_O and lyophilized.

Physiochemical properties and results of analysis with ^1^H NMR and ^13^C NMR are reported for the lead compounds **1968-1**, **1968-4**, **1968-16** and **1968-22** in the Appendix A. 

### 4.3. In Vitro Radioligand Competition Binding Assays

MOR binding assay: Rat cortices were homogenized using 50 mM Tris, pH 7.4, and centrifuged at 16,500 rpm for 10 min. The pellets were resuspended in fresh buffer and incubated at 37 °C for 30 min. Following incubation, the suspensions were centrifuged as before, the resulting pellets were resuspended in 100 volumes of 50 mM Tris, pH 7.4 plus 2 mg/mL bovine serum and the suspensions combined. Each assay tube contained 0.5 mL of membrane suspension, 2 nM [^3^H]-DAMGO, and 0.02 mg/mL mixture in a total volume of 0.65 mL [13,45,53,55].

KOR binding assay: Guinea pig cortices and cerebella were homogenized using 50 mM Tris, pH 7.4, 10 mM MgCl_2_-6H_2_O, 200 µM and centrifuged and incubated as above. Each assay tube contained 2 nM [^3^H]U69,593. Assay tubes were incubated for 2 h at 25 °C. Unlabeled U50,488 was used as a competitor to generate a standard curve and determine nonspecific binding [31,55].

DOR binding assay: Rat cortices were homogenized using 50 mM Tris, pH 7.4, 10 mM MgCl_2_-6H_2_O, 200 µM PMSF, centrifuged and incubated as above. Each assay tube contained 2 nM [^3^H]-DPDPE. Assay tubes were incubated for 2.5 h at 25 °C. Unlabeled DPDPE was used as a competitor to generate a standard curve and determine nonspecific binding [45].

All three binding assays were terminated by filtration through GF/B filters, soaked in 5 mg/mL bovine serum albumin, 50 mM Tris, pH 7.4, on a Tomtec Mach II Harvester 96. The filters were subsequently washed with 6 mL of assay buffer. Bound radioactivity was counted on a Wallac Betaplate Liquid Scintillation Counter. Experiments were conducted using two replicates and repeated twice. Values in Figure 2 represent the mean ± SD of two experiments with two replicates in each test.

### 4.4. In Vivo Testing

#### 4.4.1. Animals and Drug Administration

Adult male wild-type C57BL/6J mice weighing 20–25 g were obtained from Jackson Labs (Bar Harbor, ME, USA). Food pellets and distilled water were available ad libitum. All mice were kept on a 12 h light–dark cycle and were housed and cared for in accordance with the 2011 National Institute of Health *Guide for the Care and Use of Laboratory Animals*. All results of animal testing are reported in accordance with ARRIVE guidelines [87]. All of the procedures used herein were preapproved by the Institutional Animal Care and Use Committee at the University of Florida, under protocol #201910926.

For intracerebroventricular (i.c.v.) administration, the **1968**-series compounds were dissolved in dimethyl sulfoxide (DMSO), followed by the addition of sterile saline (0.9%) so that the final vehicle was 50% DMSO and 50% saline, and the i.c.v. injections were performed as described previously [88]. For per os (p.o.) administration, **1968-4** and **1968-22** were administered in 10% Ethanol/10%Tween-80/80% 0.9% saline. All solutions for animal administration were prepared fresh daily.

#### 4.4.2. Antinociceptive Testing

The 55 °C warm-water tail-withdrawal assay was performed in mice as previously described [13,89], with the latency of the mouse to withdraw its tail from the water taken as the endpoint (a cut-off time of 15 sec was used in this assay). Antinociception was calculated according to the following formula: % antinociception = 100 × (test latency − control latency)/(15 − control latency). Tail-withdrawal data points are the means of 8–16 mice, unless otherwise indicated, with SEM shown by error bars.

The opioid receptor involvement in the agonist activity of the heterocyclic peptidomimetics was determined by pretreating mice with a single dose of β-funaltrexamine (β-FNA, 10 mg/kg, i.p.) or nor-BNI (10 mg/kg, i.p.) 24 h in advance of administration of a dose of **1968**-series compound. Additional mice were pretreated with a single dose of naloxone (10 mg/kg, s.c.) or naltrindole (20 mg/kg, i.p.) 15 min in advance of administration of the **1968**-series compound.

To determine antagonist activity, mice were pretreated with **1968-22** 110 min prior to the administration of the MOR-preferring agonist morphine (10 mg/kg, i.p.), KOR-selective agonist U50,488 (10 mg/kg, i.p.) or DOR-selective agonist SNC-80 (100 nmol, i.c.v.); at this time, the antinociceptive activity of **1968-22** had dissipated. Antinociception produced by the established opioid agonists was then measured 40 min after their administration.

#### 4.4.3. Acute Antinociceptive Tolerance Determination

A standardized state of tolerance was induced by administration of morphine or test compound at times 0 and 8 h [54,62,90] to quantitatively evaluate development of acute opioid tolerance. This assay was used to efficiently measure the potential of compounds to cause tolerance using a minimum amount of compound, while yielding reliable results. Mice were administered an ED_50_ dose (i.c.v.) of test compound in the morning (time = 0) and a second dose (varying between 0.001–100 nmol, i.c.v.) 8 h later. The degree of tolerance was calculated from the shift in ED_50_ value from the singly to repeatedly treated condition [63,91]. All compounds were administered i.c.v., with antinociception assessed 30 min after injection of morphine or at 20 min, the time of the peak antinociceptive effect of **1968-22**, as determined in their initial antinociceptive characterization.

#### 4.4.4. Respiration and Ambulation

Respiration rates (in breaths per minute) and animal locomotive activity (as ambulations) were assessed using the Oxymax/CLAMS system (Columbus Instruments, Columbus, OH), as described previously [62,63]. Mice were habituated to their individual sealed housing chambers for 60 min before testing. Mice were administered **1968-22** (0.3–100 nmol, i.c.v.), morphine (100 nmol, i.c.v.), or vehicle, as indicated, and five min later confined to the CLAMS testing chambers. Pressure monitoring within the sealed chambers measured frequency of respiration. Infrared beams located in the floor measured locomotion as number of beam breaks. Respiration and locomotive data were averaged over 20 min periods for 120 min post-injection of the test compound. Data are presented as % vehicle response ± SEM, ambulation or breaths per minute.

#### 4.4.5. Evaluation of Potential Conditioned Place Preference and Conditioned Place Aversion

An automated, balanced three-compartment place-conditioning apparatus (San Diego Instruments, San Diego, CA, USA) and a 2-day counterbalanced place-conditioning design were used, similar to methods previously described [13,92]. The amount of time subjects spent in each of the three compartments was measured over a 30 min testing period. Prior to place conditioning, an initial preference test was performed in which the animals could freely explore all open compartments; the animals did not demonstrate significant differences in their time spent exploring the outer left (554.6 ± 17.4 s) versus right (543.1 ± 12.7 s) compartments (*p* = 0.65, Student’s *t*-test). For place conditioning, mice were administered 0.9% saline (i.p.) and consistently confined in a randomly assigned outer compartment: half of each group in the right chamber, and half in the left chamber. Four hours later, mice were administered the test compound and confined to the opposite compartment for 40 min. To determine if **1968-22** across a range of doses (0.1, 1 or 10 nmol, i.c.v.) produced CPP or CPA, mice were place conditioned in this way for two days, with a final preference test taken on the fourth day, as this has been shown to produce dependable morphine CPP and U50,488-induced CPA [80]. Additional groups of mice were place conditioned with morphine or U50,488 (100 nmol, i.c.v.) as positive controls.

### 4.5. Statistical Analysis

All dose–response lines were analyzed by regression, and ED_50_ (effective dose producing 50% antinociception) values and 95% confidence intervals (C.I.) were determined using individual data points from graded dose–response curves with Prism 8.0 software (GraphPad, La Jolla, CA, USA). Percent antinociception was used to determine within group effects and to allow comparison to baseline latency in tail-withdrawal experiments. The statistical significance of differences between ED_50_ values was determined by evaluation of the ED_50_ value shift via nonlinear regression modeling with Prism software. Significant differences in behavioral data were analyzed by ANOVA (one-way or two-way with repeated measures (RM), as appropriate). Significant results were further analyzed with Sidak’s, Tukey’s, or Dunnett’s multiple comparison post hoc tests, as appropriate. Data for conditioned place preference experiments were analyzed by two-way RM ANOVA, with analyses examining the main effect of the conditioned place preference phase (e.g., pre- or post-conditioning) and the interaction of drug pretreatment. Significant effects were further analyzed using Sidak’s HSD post hoc testing. All data are presented as mean ± SEM, with significance set at *p* < 0.05.

## Data Availability

All data needed to evaluate the conclusions in the manuscript are present in the manuscript and Appendix A. Data related to this manuscript may be requested from the authors.

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
