# Peer review of "Bis-Cyclic Guanidine Heterocyclic Peptidomimetics as Opioid Ligands with Mixed μ-, κ- and δ-Opioid Receptor Interactions: A Potential Approach to Novel Analgesics"

_ijms, 2022, doi:10.3390/ijms23179623_

Round 1

Reviewer 1 Report

This paper discusses design and synthesis of a bis-cyclic guanidine peptidomimetics and in vitro and in vivo characterization of their opioid activity. The goal was to develop ligands with mixed opioid receptor interactions exhibiting fewer side effects. The data from in vitro and in vivo screening suggest that analog 1968-22 may hold potential for a new and safer type of analgesic.

Overall, this paper is well written and deals with an important and timely and ongoing problem in the field.

A few things to consider.

1.       Line 12, Abstract

“…. receptor interactions is theorized ….” Suggested maybe? Theorized sounds as not supported by evidence, and some recently published paper on this topic are cited in the introduction.

2.       Scheme 1 can be improved by defining diversity better (R1-R3 side chains description)

3.       Line 123-128

The choice of amino acids such as Phe, Tyr, and Tyr analog (Dmt), in design of a library based on a tripeptide core is well explain. However, the use of fourth amino acid Ala at position R1 and Leu at positions R2 and R3 is less obvious. Please elaborate on this.

4.       Line 131-132

“Some of the compounds were also active against kappa and/or delta receptors.  

Be more specific, identify in the text compounds with activity for kappa or delta and their respective Ki values.

5.       Scheme 2

What does exact mass mean? A calculated expected value? Reference the SI section for compounds full characterization.

Why are the NMR and MS values only given for four compounds? All compounds referenced on Scheme 2 should be fully characterized. An actual NMR spectrum in addition to peak assignation should be included. What type of MS was used for characterization? Is it LC-MS or MALDI? Use proper reporting for the MS values, m/z, calcd, found

6.       Line 156

While, should be a lower case

7.       Figure 1-8

Figures should be a better quality. Figure 3 is missing.

8.       Line 409-410

“The purity of the crude samples was estimated based on the UV traces recorded.”

Please clarify.

9.       Line 559.

Please enter the SI material provided with the manuscript.
